# Control of the neuroprotective Lipocalin Apolipoprotein D expression by alternative promoter regions and differentially expressed mRNA 5′ UTR variants

Sergio Diez-Hermano[1,2], Andres Mejias[3], Diego Sanchez[1], Gabriel Gutierrez[3⊙], Maria D. Ganfornina[1⊙]*

1 Departamento de Bioquimica y Biologia Molecular y Fisiologia, Instituto de Biologia y Genetica Molecular, Universidad de Valladolid-CSIC, Valladolid, Spain, 2 Departamento de Matemática Aplicada, Universidad Complutense, Madrid, Spain, 3 Departamento de Genetica, Universidad de Sevilla, Sevilla, Spain

⊙ These authors contributed equally to this work.
* opabinia@ibgm.uva.es

**Data Availability Statement:** All relevant data are within the paper and its Supporting Information files.

## Abstract

The Lipocalin Apolipoprotein D (ApoD) is one of the few genes consistently overexpressed in the aging brain, and in most neurodegenerative and psychiatric diseases. Its functions include metabolism regulation, myelin management, neuroprotection, and longevity regulation. Knowledge of endogenous regulatory mechanisms controlling brain disease-triggered ApoD expression is relevant if we want to boost pharmacologically its neuroprotecting potential. In addition to classical transcriptional control, Lipocalins have a remarkable variability in mRNA 5′UTR-dependent translation efficiency. Using bioinformatic analyses, we uncover strong selective pressures preserving ApoD 5′UTR properties, indicating unexpected functional conservation. PCR amplifications demonstrate the production of five 5′UTR variants (A-E) in mouse ApoD, with diverse expression levels across tissues and developmental stages. Importantly, Variant E is specifically expressed in the oxidative stress-challenged brain. Predictive analyses of 5′UTR secondary structures and enrichment in elements restraining translation, point to Variant E as a tight regulator of ApoD expression. We find two genomic regions conserved in human and mouse ApoD: a canonical (α) promoter region and a previously unknown region upstream of Variant E that could function as an alternative mouse promoter (β). Luciferase assays demonstrate that both α and β promoter regions can drive expression in cultured mouse astrocytes, and that Promoter β activity responds proportionally to incremental doses of the oxidative stress generator Paraquat. We postulate that Promoter β works in association with Variant E 5′UTR as a regulatory tandem that organizes ApoD gene expression in the nervous system in response to oxidative stress, the most common factor in aging and neurodegeneration.

## Introduction

Lipocalins are a family of extracellular proteins that can bind small hydrophobic ligands by means of a highly conserved β-barrel tertiary structure [1]. This protein family shows

**Funding:** M.D.G. adn D.S. BFU2015-68149-R Ministerio de Ciencia e Innovacion http://www.ciencia.gob.es/ The funders had no role in study design, data collection and analysis, decision to publish, or preparation of the manuscript.

**Competing interests:** The authors have declared that no competing interests exist.

moonlighting properties and a wide functional diversity [2]. Only a few Lipocalins are expressed within the nervous system: Apolipoprotein D (ApoD), mainly in oligodendrocytes and Schwann cells and in astrocytes; Lipocalin-type prostaglandin D synthase (Ptgds), in oligodendrocytes; or Lipocalin 2 (Lcn2), expressed by reactive astrocytes [1,3]. In addition to classical transcriptional control, we have recently reported that Lipocalins have a remarkable variability in 5' UTR-dependent translation efficiency, which explains differences in global protein abundance depending on their evolutionary history [4]. These differences in post-transcriptional regulation set apart a group of mammalian Lipocalins, where ApoD stands as the earliest diverging chordate protein according to a congruent molecular phylogeny based on primary sequence and gene architecture [5].

ApoD gene is expressed in several mammalian tissues, and the protein was first detected complexed to blood circulating lipoproteins. However, the nervous system is a prominent organ of expression, particularly in rodents [reviewed by 6, 7]. Particularly intriguing is the fact that ApoD is one of the few genes consistently over-expressed in the aging brain of all vertebrate species tested so far [8]. Moreover, ApoD expression is boosted in an amazingly wide array of neurodegenerative and psychiatric diseases of diverse etiology, including schizophrenia and bipolar disorder [9], Alzheimer´s disease [10], demyelinating diseases like Multiple Sclerosis [11] or lysosomal storage diseases, like Niemann Pick type C disease [12]. Oxidative stress appears clearly as a common factor for brain aging and diseases of such varied etiologies. The mouse, as model organism, has yielded a wealth of knowledge about functions of this atypical apolipoprotein: it delays brain aging [13], promotes longevity and dopaminergic neuron survival under oxidative stress conditions [14,15], and controls myelin phagocytosis and restrains inflammation after peripheral nerve injury, thus accelerating recovery [16,17]. Finally, ApoD is required for myelin compaction during normal development [18]. How are all these functions coordinated? What are the mechanism controlling the upregulation of ApoD upon a neural insult due to trauma, exposure to exogenous toxics, or a wide array of neurodegenerative processes?

The influence of ApoD on the transcriptome response to Paraquat-induced oxidative stress has been characterized in the cerebellum [19], one of the brain regions particularly vulnerable to this insult [20]. On the other hand, ApoD expression is transcriptionally regulated by the JNK pathway in human astrocytoma cells upon oxidative stress [15], and by the MEK/ERK pathway in mouse fibroblasts (NIH/3T3 cells) upon growth arrest [21]. However, other layers of ApoD expression control have not been explored.

The postranscriptional regulatory control of gene expression by upstream untranslated regions (5' UTR) of mature mRNAs has gained attention in recent years [reviewed by 22, 23]. They are key components of the post-transcriptional regulation due to their impact on translation efficiency. 5' UTR features such as length, G+C content, secondary structure, as well as the presence of particular sequence motifs have been demonstrated to impact on gene translation levels [24,25].

In this work, we study both *in silico* and experimentally the role of mRNA 5' UTR sequences and gene promoter regions on the regulation of mouse Lipocalin ApoD gene expression in different tissues, developmental stages and physiological conditions.

## Methods

### Computational analyses

Mammalian ApoD 5' UTR sequences were obtained from the AceView database [26]. Only annotated transcripts containing an ApoD coding sequence (CDS) coincident with NCBI

Reference Sequence (RefSeq Database) were selected. Nucleotide sequences obtained from AceView were confirmed in ASPIcDB [27], which also allowed including alternative transcripts.

UTR regions were analyzed with EMBOSS Infoseq, Dreg and Getorf tools [28]. Repetitive motifs were located with Repeatmasker (http://repeatmasker.org). The RNAshape algorithm (http://bibiserv.techfak.uni-bielefeld.de/rnashapes) [29] was employed to predict the minimal folding energy (MFE) and the suboptimal structures of ApoD 5' UTRs, selecting a range of MFE + 5 Kcal/mol for the suboptimal structures. We evaluated structural similarities of the predicted alternative UTR structures with RNAforester (http://bibiserv2.cebitec.uni-bielefeld.de/rnaforester) [30], and the structures were studied with PseudoViewer [31]. To predict regulatory motifs in 5' UTR we used Predict a motif [32], the RNAalifold algorithm [33] and RNAstructure (v6.1) [34].

AceView database annotations were used to map exon-intron organization. 5' UTR genomic regions were additionally examined with ExonScan [35] to predict potential exons. The presence and category of constitutive, alternative or cryptic splicing sites flanking exons were predicted with ASSP [36].

Promoter regions were identified as those annotated by the ENCODE project [37], and predicted by the Genomatix database (http://www.genomatix.de). Promoter predictions were carried out by NNPP (http://www.fruitfly.org/seq_tools/promoter.html) [38], FPROM (http://www.softberry.com/berry.phtml?topic=fprom&group=programs&subgroup=promoter) [39], YAPP (http://www.bioinformatics.org/yapp/cgi-bin/yapp.cgi) [40] and Promoter 2.0 (http://www.cbs.dtu.dk/services/Promoter/) [41] algorithms. Promoter predictions and ApoD gene structure were visualized with the IGV browser V2.5.3 (https://software.broadinstitute.org/software/igv) [42]. To find internal duplications in the 5' upstream genomic regions of human and mouse ApoD we used PLALIGN [43]. In order to find possible regulatory sites in ApoD promoter regions, we performed a computational sequence search for potential transcription factor binding sites using ModelInspector (http://www.genomatix.de) [44].

## Animals and cell cultures

C57BL/6J mice (RRID:IMSR_JAX:000664) were maintained in positive pressure-ventilated racks at 25±1°C with 12 h light/dark cycle, fed ad libitum with standard rodent pellet diet (Global Diet 2014; Harlan Inc., Indianapolis, IN, USA), and allowed free access to filtered and UV-irradiated water. Mice were normally housed in groups of 3–4 animals/cage, but were kept individually caged for the experimental treatment. The University of Valladolid Animal Care and Use Committee following the regulations of the Care and the Use of Mammals in Research (European Commission Directive 86/609/CEE, Spanish Royal Decree ECC/566/2015) approved experimental procedures (CEEBA Univ. Valladolid, project #8702359). For oxidative stress treatment, six month old male mice were randomly subject to either a single intraperitoneal injection of Paraquat (PQ, Sigma; 30 mg/kg) in 200 μl sterile saline (experimental group, n = 6), or a similar volume of sterile saline (control group, n = 4). Six hours after injections, mice were euthanized with $CO_2$ and their cerebella immediately removed and frozen. No animal suffering was observed during the short treatment period. Other tissues (adipose, heart, colon and lung) were extracted from control mice. Whole brain or cerebellum were extracted from embryos (E13.5) or postnatal control mice (P10) respectively (n = 3/stage), euthanized with $CO_2$ and their tissues immediately frozen.

The mouse astrocytic cell line IMA2.1 (RRID:CVCL_X370) was grown in Dulbecco Modified Eagle´s Medium (DMEM) without phenol red, with 5% heat-inactivated fetal bovine serum (FBS), 2 mM L-glutamine, 100 U/ml penicillin, 100 U/ml streptomycin, and 0.25 μg/ml

amphoterycin, with subculture cycles every 48 hours when they reach 80% confluence. Oxidative stress treatment of cells (0.5 or 1 mM PQ) was carried out in low serum media (0.2% FBS; all other components as above).

## Immunocytochemistry

Cultured IMA2.1 astrocytes attached to poly-L-lysine (Sigma)-treated coverslips were fixed with 4% formaldehyde, washed in phosphate buffered saline (PBS), blocked and permeabilized with Tween-20 (0.1%) and 1% non-immune calf serum. We used a goat polyclonal anti-mouse ApoD (SC Biotechnology) as primary antibody, and Alexa 488-conjugated donkey anti-goat IgG serum (Jackson Immunoresearch) as secondary antibody. Coverslips were mounted with EverBrite™-DAPI Mounting medium, and sealed with CoverGrip™ sealant (Biotium). Cells were visualized and photographed with an Eclipse 90i (Nikon) fluorescence microscope equipped with a DS-Ri1 (Nikon) digital camera, and images were processed and analysed with the Fiji Program.

## Genomic PCR, RT-PCR and Real-time quantitative PCR

Mouse tissues used for mRNA expression studies were stored at -80˚C, and RNA was extracted using QIAzol Lysis Reagent (Qiagen). RNA concentration was measured with a Nanodrop spectrophotometer, and its quality assessed by 260/230 and 260/280 spectrophotometric ratios measured with a spectrophotometer and by agarose electrophoresis. RNA obtained from individual samples of the same tissue or experimental condition were pooled in equimolar amounts to be reversed transcribed. Following DNAse treatment, 500 ng of total RNA were reverse-transcribed with PrimeScript (Takara Bio Inc., Otsu, Japan) using Oligo-dT primers and random hexamers. Genomic DNA absence was also confirmed by RT(-) amplifications. Genomic DNA was obtained from a mouse brain after RNA purification following the QIAzol (Qiagen) protocol, and its concentration was measured with Nanodrop.

Mouse cDNA was used as a template for standard RT-PCR using GoTaq® (Promega), or quantitative real-time RT-PCR (RT-qPCR) using SybrGreen (SYBR® Premix Ex Taq™ kit, Takara). Genomic DNA was used for testing 5' UTR primers, and for amplification and cloning of ApoD promoters. The primers used in PCR amplifications are shown in Table 1.

The mRNA transcription levels were assessed with the ΔΔCT method [45] using normalization to the Rpl18 gene for each condition. Five technical replicas were performed with each sample set. Bacterial plasmid DNA was used as negative template control. Statistically significant differences of gene transcriptional changes were evaluated with a Mann-Whitney U-test [46] using ΔCT of each replica (calculated by subtracting the average CT of the reference gene for each sample).

## Promoter cloning and gene expression evaluation

The two mouse ApoD genomic regions showing sequence conservation with human ApoD sequences (Fig 5) were PCR-amplified and cloned in the pCR®II-TOPO® vector (Invitrogen), and confirmed by DNA sequencing. These proposed promoter regions (α and β) were then digested with restriction enzymes and directionally cloned in the pGL4.10[*luc2*] reporter vector (Promega). Astrocyte IMA2.1 cells were Lipofectamine (Invitrogen)-transfected with the mouse ApoD promoter (α or β)-pGL4.10[*luc2*] plasmids. The promoter-driven expression of Luciferase was tested with the Dual-Luciferase® Reporter Assay System (Promega) following the manufacturer's specifications. Briefly, cells were transfected with each luciferase reporter construct and a Renilla expression vector at a 10:1 ratio. After an 18 h post-transfection period, cells were incubated for 2 h with Low Serum media (control conditions) or PQ (0.5 or 1 mM)

**Table 1. PCR primers used in this work.**

| Experiment | Primer | Sequence |
|---|---|---|
| Genomic PCR; RT-PCR | pU1-F | AGGGGACAGACACAGCATCCCA |
| Genomic PCR; RT-PCR | pU2-F | GGAGGATTCTGGGTGGAAACTTCAG |
| Genomic PCR; RT-PCR/qPCR | pU3-F | AGTTGGAGCTTGCACTTGGGGT |
| Genomic PCR; RT-PCR/qPCR | pU4-F | CCTCGGTGCTGAGGAGAATTCCA |
| Genomic PCR; RT-PCR/qPCR | pU2-R | AGCCTTCAGTTGGTGCTCACTGT |
| Genomic PCR; RT-PCR/qPCR | CDS-R | CGTGGCCAGGAACATCAGCATG |
| RT-PCR | 1-F | GAAGCCAAACAGAGCAACG |
| RT-PCR | 1-R | AGCCTCACAGACTGATTCAGGG |
| RT-PCR | 2-R | AGCACTTCGATGTTTCCGTTCTCC |
| RT-PCR | 3-R | AGCTTGGCTGGCTCTGAGACG |
| RT-PCR | 4-R | TGTTTCTGGAGGGAGATAAGGA |
| Promoter Cloning | PROMA-F | GGAACGTTCAGCAGATCACTT |
| Promoter Cloning | PROMA-R | GAGAGCGAGAGCGAGAGAGAAAGAC |
| Promoter Cloning | PROMB-F | TGTTATTGGAACCCGTTTTCAGGTG |
| Promoter Cloning | PROMB-R | ACCTCTTTTCAAGCATCTCTTGTTGG |
| RT-qPCR | Rpl18-F | TTCCGTCTTTCCGGACCT |
| RT-qPCR | Rpl18-R | TCGGCTCATGAACAACCTCT |

in Low Serum media. Luminescence was then measured in cell lysates with a BMG LABTECH 96 microplate luminometer. Experiments were performed in triplicates. Promoter activities were expressed relative to Renilla activity. PQ-dependent activity was normalized to values obtained in control conditions.

## Statistical analysis

Statistical analyses were performed with SPSS v.19 (IBM) and SigmaPlot v.11.0 (Systat). A $p$ value $< 0.05$ was considered as threshold for significant changes and marked with asterisks in figures. The tests used for each experiment are described in figure legends.

# Results

## ApoD 5' UTR properties evidence selective pressure and tight translational control

The particular position of ApoD as an early-diverging Lipocalin within chordate evolution [5], its functional diversity and apparent pleiotropy (influencing a wide array of processes, from metabolic traits to myelin management, neurodegeneration or aging progression), together with its tissue and stimulus-specific expression, triggered the present analysis of regulatory regions previously unexplored for the mouse ApoD gene.

First we focus on the *in silico* analysis of ApoD 5' UTR in mammals, as a potential source of post-transcriptional variations in its gene expression control. A comparison of the 5' UTR sequences of ApoD mRNAs of several mammalian orders (using the transcripts annotated as RefSeq) shows a 78% identity [4], a value comparable to those obtained when comparing the 3rd position of codons in the CDS of the same set of transcripts. This strong sequence conservation is apparent in multiple sequence alignment of sequences diverging up to 120 My ago (Fig 1).

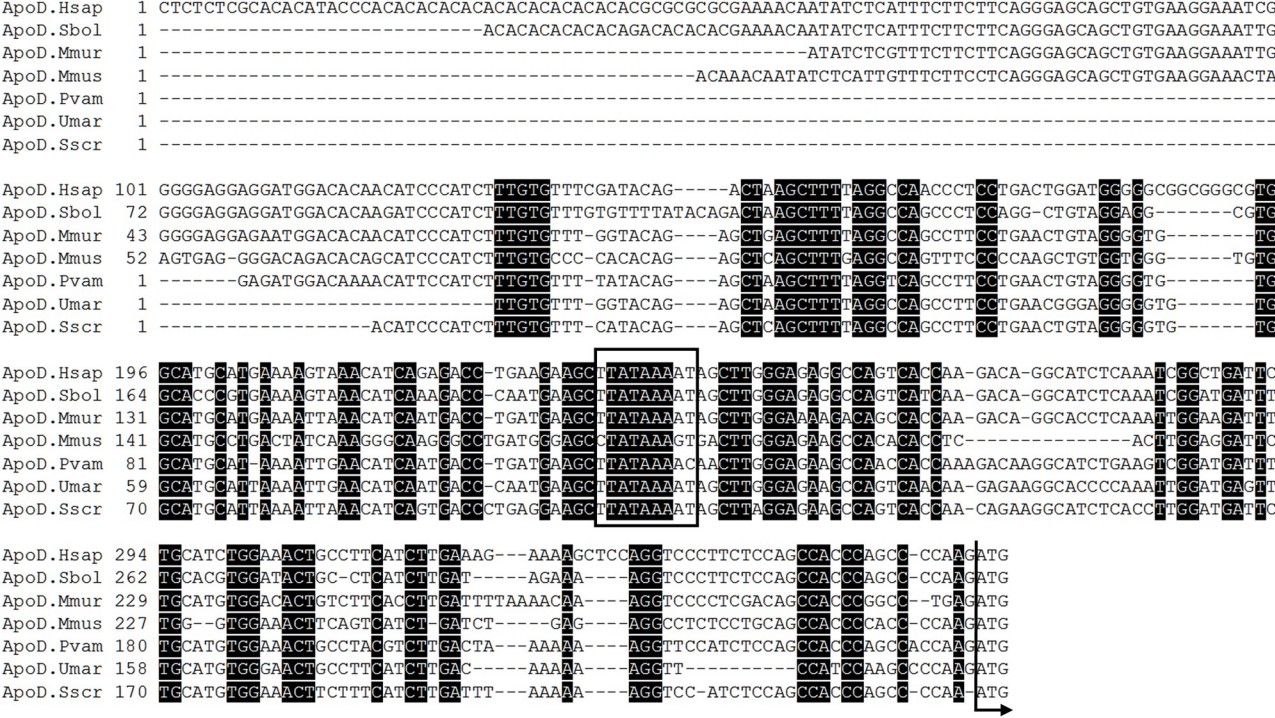

**Fig 1. Evolutionary conservation of ApoD 5' UTR in mammals.** Multiple sequence alignment of the RefSeq 5' UTRs of several mammalian species. Nucleotide sequence identity across all taxa are highlighted in black. A conserved TATA box is marked by a square, and the beginning of the CDS is pointed by an arrow. Hsap: Homo sapiens (primates); Sbol: Saimiri boliviensis (primates); Mmur: Microcebus murinus (primates); Mus musculus (rodentia); Pvam: Pteropus vampyrus (chiroptera); Umar: Ursus maritimus (carnivora); Sscr: Sus scrofa (artiodactyla).

This high sequence conservation for an early-diverging chordate Lipocalin suggests the existence of strong selective pressure on ApoD 5' UTR sequence and, therefore, on its regulatory role in ApoD expression.

Properties of 5' UTRs that are suitable to promote gene translation are a short sequence length, a reduced G+C content, a negative minimum folding energy (MFE) for their secondary structure, and the absence of unstructured (linear) regulatory elements such as upstream initiation codons (uAUG) or of upstream open reading frames (uORF), and [25,47,48,49]. When analyzing these properties in human and mouse ApoD 5' UTRs, we find that although they show lengths and G+C content values similar to their average in both species, the presence of uAUGs and uORFs [4] suggests a tight translational control of this gene.

## Alternative 5' UTRs in mouse ApoD are differentially expressed depending on tissue or physiological conditions

The accumulated knowledge on ApoD function in the mouse model, particularly relevant to understand its role in nervous system maintenance and resistance to neurodegeneration, led us to experimentally test the role of mouse ApoD 5' UTR and to analyze its regulatory potential.

An analysis of the genomic sequence of the 5' UTR region of human and mouse ApoD reveals that it is composed of several exons, an arrangement found in all mammals studied so far [4]. Fig 2A shows the exon arrangement of the mouse ApoD gene, where four exon fragments contribute to the 5' UTR as predicted by ExonScan (Fig 2B) and ASSP (Fig 2C). These

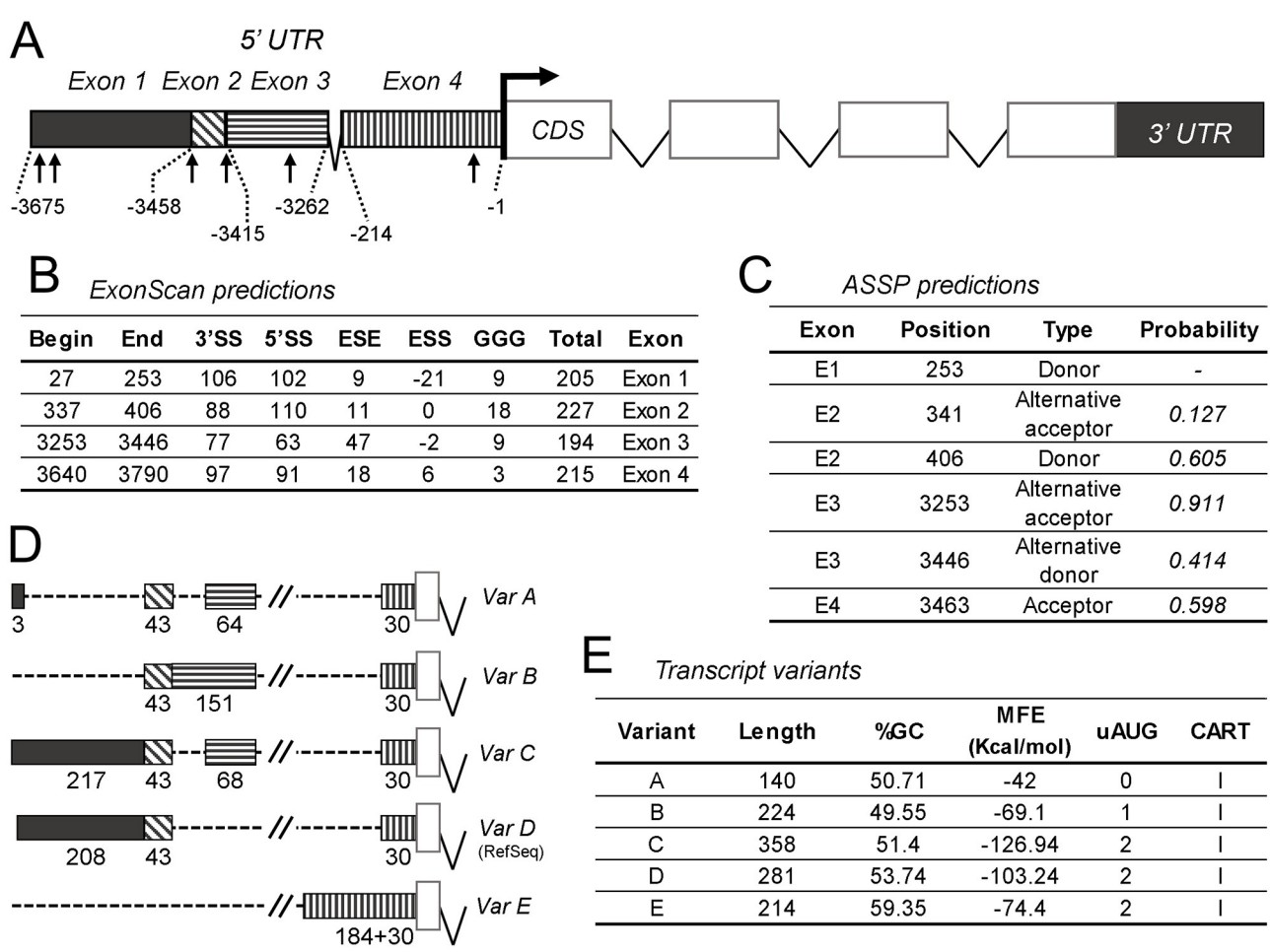

**Fig 2. Bioinformatic analyses of mouse ApoD 5' UTR region.** A) Predicted exonic structure of mouse ApoD 5' UTR region. Arrows point to predicted splice sites. Exon boundaries are marked by numbers in relation to the ORF. B) ExonScan prediction of potential exonic 5' UTR sequences. 5'SS (donor) and 3'SS (acceptor) indicate probabilities of splice sites in comparison to decoy at both ends. ESE (exonic splicing enhancers) values are the estimated odd of a given string being an exon relative to being an intron. ESS (exonic splicing silencers) values are the estimated odd of being a pseudoexon in contrast to being an exon. Intronic GGG takes a small constant value when located 100–40 bases upstream or 10–70 bases downstream a candidate exon. Total refers to the sum of scores for every candidate exon. C) ASSP classification and estimated probability of predicted exonic 5' UTR sequences. Both ExonScan and ASSP predictions are very similar. D) Exon composition of the transcriptional variants of mouse ApoD 5' UTR annotated in AceView database. E) Primary and secondary structure properties of mouse ApoD 5' UTR variants, including estimated minimal folding energy (MFE), number of upstream AUG codon (uAUG) and class resulting from the classification and regression tree (CART) method.

exons contribute totally or partially to the set of 5' UTR variants (Fig 2D) found by *in silico* analysis (AceView and ASPIcDB). Such a set of five variants is the largest number of alternative 5' UTRs predicted for a mammalian Lipocalin gene [4]. The presence of different numbers and fragments of 5' UTR exons in the variants suggests RNA splicing as a mechanism to generate such diversity. Predicted splice sites are pointed by arrows in Fig 2A. Fig 2E shows several properties of each variant, all of which categorize them as "low translation efficiency" 5' UTRs, based on the classification and regression tree (CART, class I) method [50]. This circumstance also suggests that different cell types or physiological states of a given cell could influence the regulatory impact of ApoD 5' UTR depending on the variant expressed.

To confirm experimentally the expression of the predicted 5' UTR mouse variants we used RT-PCR amplifications with a combination of oligonucleotide pairs (Fig 3A and Table 1) that

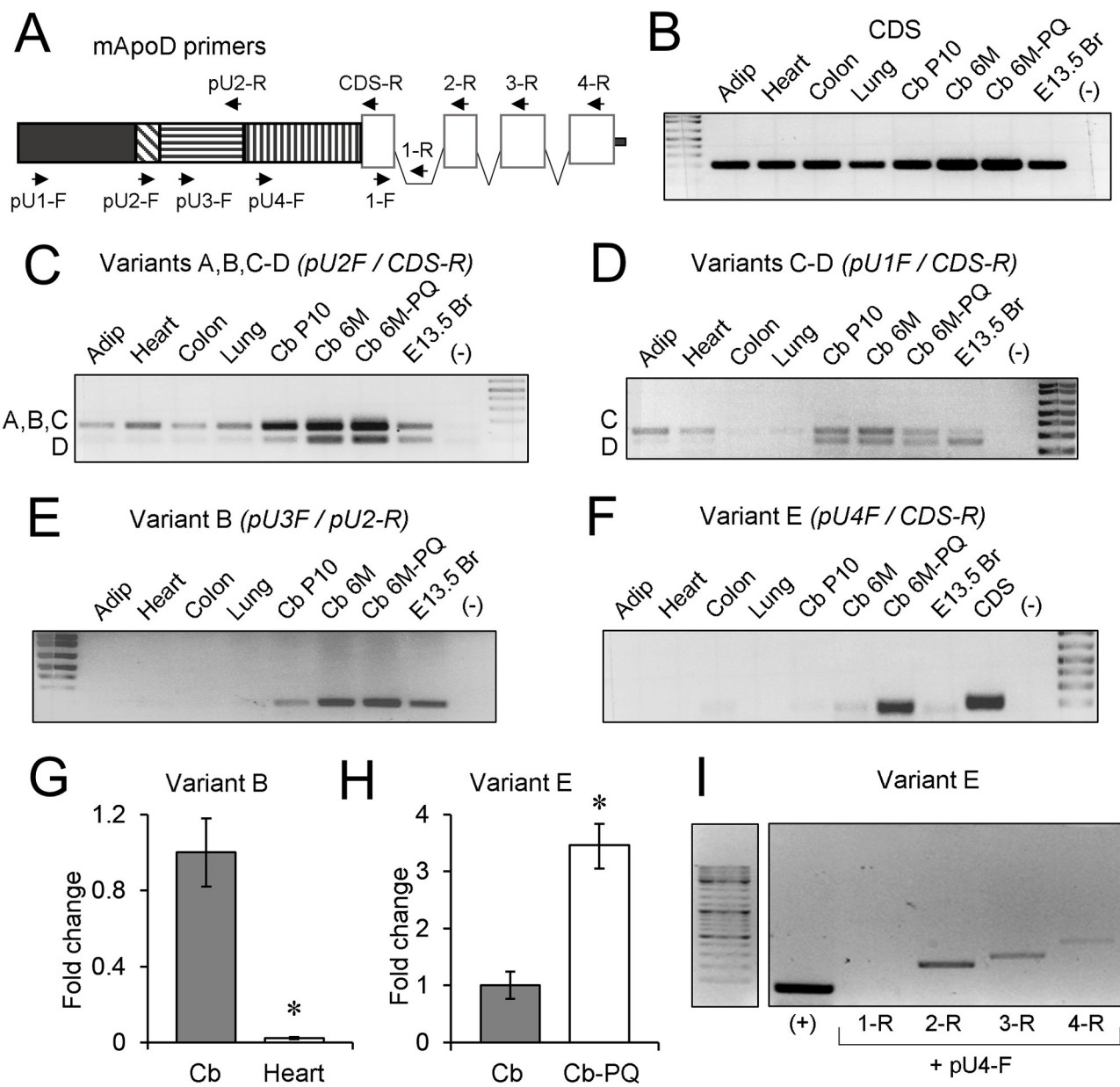

**Fig 3. Experimental demonstration of tissue, time and condition-dependent expression of mouse ApoD 5' UTR variants.** A) Schematic diagram of 5' UTR and CDS exons of mouse ApoD to map the location and direction of primers used in RT-PCR and RT-qPCR experiments. B) Control RT-PCR amplifications of mouse ApoD CDS in several tissues. Adip: adipose tissue; Cb: cerebellum; P10: postnatal day 10; 6M: six months old; E13.5 Br: brain of 13.5 days mouse embryo; PQ: Paraquat, 6 h in vivo treatment; (-): negative control template. Biological replicas used in these experiments are detailed in the methods section. C-F) PCR amplification products of mouse ApoD 5' UTR variants. Primer combination and/or band size help us to discriminate among variants. G-H) Real time RT-qPCR expression levels of variants B and E demonstrate brain specificity of variant B and PQ induction of variant E. Average fold change and standard deviation of five technical replicas are represented. Significance was assessed on the original ΔCT values of each replica with a Mann-Whitney U-test. I) RT-PCR demonstration of full transcription of ApoD mRNA containing the complete CDS with 5' UTR Variant E in cerebellum.

help us to reveal the different 5' UTR variants. We first checked that each oligonucleotide pair amplifies the expected size product from genomic DNA (not shown). We then chose cDNAs from heart, colon, lung, adipose tissue, and cerebellum as templates for amplification. To test for the existence of developmental regulation of ApoD expression within the nervous system, we compared embryonic day 13.5 brain, postnatal day 10 cerebellum, and 6 months old cerebellum. Finally, we also tested whether oxidative stress can modulate 5' UTR variants expression by comparing adult cerebellum of control mice with those treated for 6 h with Paraquat (PQ) as an oxidative stress generator. Cerebellum was selected because of its reported expression of ApoD and its sensitivity to PQ [19].

First we tested by RT-PCR amplifications that ApoD coding sequence is detected in all tissues and conditions studied (CDS; Fig 3B). Some 5' UTR variants are amplified more prominently than others when comparing with the CDS expression in each tissue or condition (Fig 3C–3F). The fragments amplified from variants A or C appear to be generally expressed in the tissues studied (Fig 3C and 3D). However, variants B, D, and E show a clear nervous system expression specificity, with minor or no presence in the other tissues tested (Fig 3C–3F). Finally, variant E shows an interesting pattern, as it appears differentially expressed in the oxidative stress-challenged cerebellum (Fig 3F).

Since the results above were obtained from standard RT-PCR evaluated after 35 cycles of amplification, we wanted to confirm our results using real-time RT-qPCR. Template concentration was evaluated by using the ΔΔCT method [45] and normalization with the Rpl18 gene. RT-qPCR confirms that variant B is more abundant in cerebellum than heart (Fig 3G), and variant E is 3-fold overexpressed in the Paraquat-treated cerebellum (Fig 3H). Since variant E has such a condition-specific expression, we wanted to confirm whether it forms part of a fully translatable mRNA containing the full ApoD CDS (all previous amplifications were carried out with a reverse primer located at the 5' end of the CDS; Primer CDS-R, Fig 3A). Using a forward primer in the first CDS exon of ApoD gene and reverse primers in different locations of ApoD CDS (coding exons 2, 3, and 4), and a negative control primer located in the first intron interrupting the CDS (primer 1-R), we were able to amplify the expected band sizes (Fig 3I).

In summary, our experiments demonstrate the existence of alternative 5' UTRs in the ApoD gene that have differential tissue or condition-dependent expression. Three of the variants are enriched particularly in the nervous system, with expression ranging from embryonic development to adulthood, and one of them (variant E) shows a remarkable specific induction upon PQ treatment.

## ApoD 5' UTR secondary structure predictions reveal varied degrees of protein expression control

The secondary structure of UTR regions is known to influence their role in gene expression regulation [25]. The 5' UTR folding can be predicted based on calculations of its free energy with the RNAshape algorithm [29], resulting in a minimal folding energy (MFE) structure as well as suboptimal structures predicted within a certain range of free energy.

The interesting functional implications of 5' UTR variants of mouse ApoD led us to study the MFE and MFE+5 Kcal/mol suboptimal structures in all variants described above. The 5' UTR modeled secondary structures of Lipocalins show limited deviations from their MFE structures [4], suggesting that MFE is representative of the native structure. This result also points to structural elements predicted to be evolutionarily conserved and potentially important for gene expression regulation. Fig 4A displays the 5' UTR MFE structures of four mammalian orthologs of mouse ApoD. Highlighted in color we show several secondary structure motifs predicted by Predict a motif [32]. These motifs are well conserved in primates (*Homo*

*sapiens* and *Saimiri boliviensis*), cow and pig sequences (*Bos taurus* and *Sus scrofa*), while the mouse orthologous 5' UTR preserves a single structural motif. Fig 4B shows the mouse ApoD RefSeq 5' UTR, highlighting in red its strong conservation with predicted suboptimal structures, making the MFE structure a good predictor of the native 5' UTR structure. Similar results are achieved when 5' UTR foldings are predicted with RNAalifold [33] and RNAstructure [34] (not shown).

Next we analyzed the secondary structures predicted for mouse ApoD alternative 5' UTRs. They can be grouped in three categories in agreement with their lengths: variants A-B, variants C-D and the unique variant E. Variant A shows the highest free energy (-42 Kcal/mol; Fig 2E), close to the -30 Kcal/mol free energy limit to negatively regulate translation [51]. Instead, the remaining variants, with low free energies (-69 to -126 Kcal/mol), are expected to inhibit gene translation more efficiently, thus representing a mechanism of restraint for ApoD protein expression.

Another interesting finding is that unstructured regulatory elements such as upstream open reading frames (uORF) are predicted only in variants B-E of ApoD 5' UTRs (highlighted in blue tones in Fig 4C), which might underlie the different degree of expression observed in several mouse tissues (Fig 3C–3F). Variant A shows a global expression in the tissues explored, coincident with its high MFE value and the lack of uORFs, both properties implicated in negative regulation of translation. The localization of uORFs, and specially their uAUGs (arrows in Fig 4C), in predicted hairpin structures of variants B-E suggests a tight translational control that could explain the restricted expression in different tissues and/or physiological conditions demonstrated above.

## Two promoter regions, differentially regulated by oxidative stress, drive the expression of mouse ApoD

The regulation of ApoD gene expression has been studied in detail by analyzing experimentally the promoter region of human ApoD [21,52,53]. A region of ~2 kb upstream of human ApoD exon 1 was reported to contain regulatory elements, such as an alternating purine-pyrimidine stretch and serum-responsive elements (SRE), that regulate ApoD expression upon a metabolic insult (serum deprivation) [52].

However, a promoter analysis of the mouse ApoD gene has not been reported so far. Using the sequence of the ~7 kb stretching from the 5'-flanking region of the human promoter region studied by Do Carmo et al. [52], we BLAST searched the 5' upstream genomic region of the mouse ApoD gene. Interestingly, two long sections showed significant sequence similarity (labeled 1 & 2 in the Dot plot shown in Fig 5A) between the human and mouse 5' upstream regions. Section 1 corresponds to part of the human promoter region and the first exon, and matches the first three 5' UTR exons of mouse ApoD. Section 2 lies in an intronic region between 5' UTR exons of both human and murine ApoD. Because of the unexpected similarity found in an intronic sequence, we first confirmed *in silico* that this region does not relate in any strand or frame to a different gene. We then searched and compared the predicted transcription initiation sites in the human gene (Fig 5B) with those of the mouse genomic sequence under study (Fig 5C). Three putative initiation sites were predicted by NNPP in the genomic 5'-flanking region of mouse ApoD gene, but only one canonical TATA box was identified in site #2 (black arrow in Fig 5C).

Two initiation sites locate in front of the 5' UTR exons that could correspond to a canonical (α) promoter homologous to the one that has been analyzed for human ApoD. Another potential site is predicted with high score in the intronic sequence downstream 5' UTR exons 1–3. These predictions were supported by other algorithms (YAPP, FPROM and Promoter 2.0; Fig

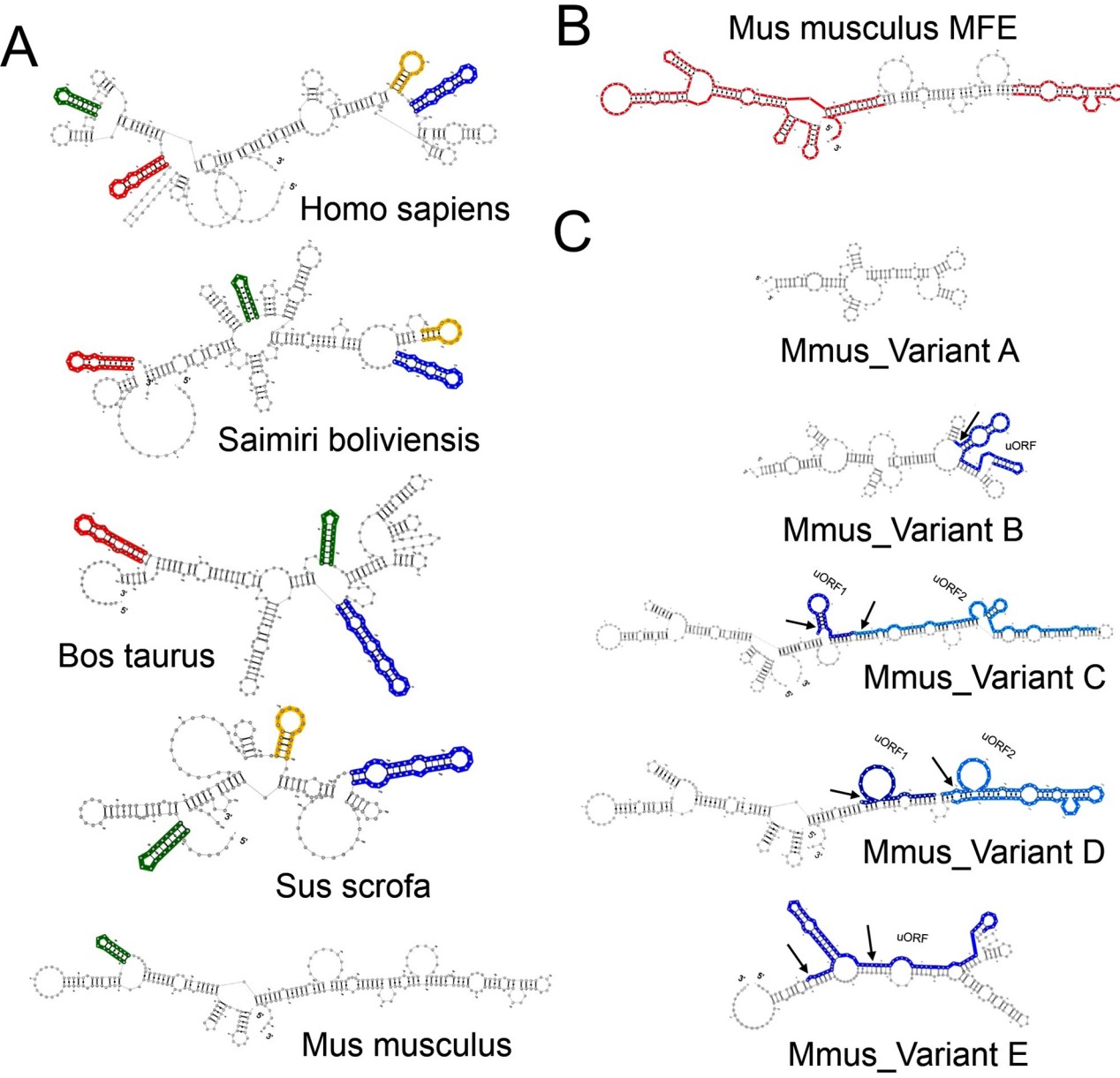

**Fig 4. Secondary structure prediction of ApoD 5' UTRs.** A) Secondary structure with minimal folding energy (MFE) predicted by the RNAshape algorithm for the RefSeq 5' UTRs of several mammalian species. Colored motifs predicted for the human 5' UTR are conserved in other mammalian sequences. B) MFE secondary structure of the ApoD mouse RefSeq 5' UTR with red-colored regions preserved in suboptimal structures (in the interval MFE + 5Kcal/mol). C) MFE secondary structure predictions of the five 5' UTR alternative variants of mouse ApoD. Blue and light-blue regions highlight predicted upstream open reading frames (uORF), while arrows point to predicted upstream initiation codons (uAUG).

6A) [39,40,41], and suggest that this region might be functioning as an alternative (β) promoter for the mouse ApoD gene (Fig 6B), located upstream of 5' UTR exon 4 (the one fully present in variant E). Likewise, the Genomatix portal predicts two promoter regions within α and β fragments (GXP_419688 and GXP_3072215; Fig 6A). Using these sequence regions, the ModelInspector algorithm [54] predicts a series of binding sites that appear enriched for a number of transcription factors (Fig 6C). Promoter α is enriched for AP1 and NFκB, which

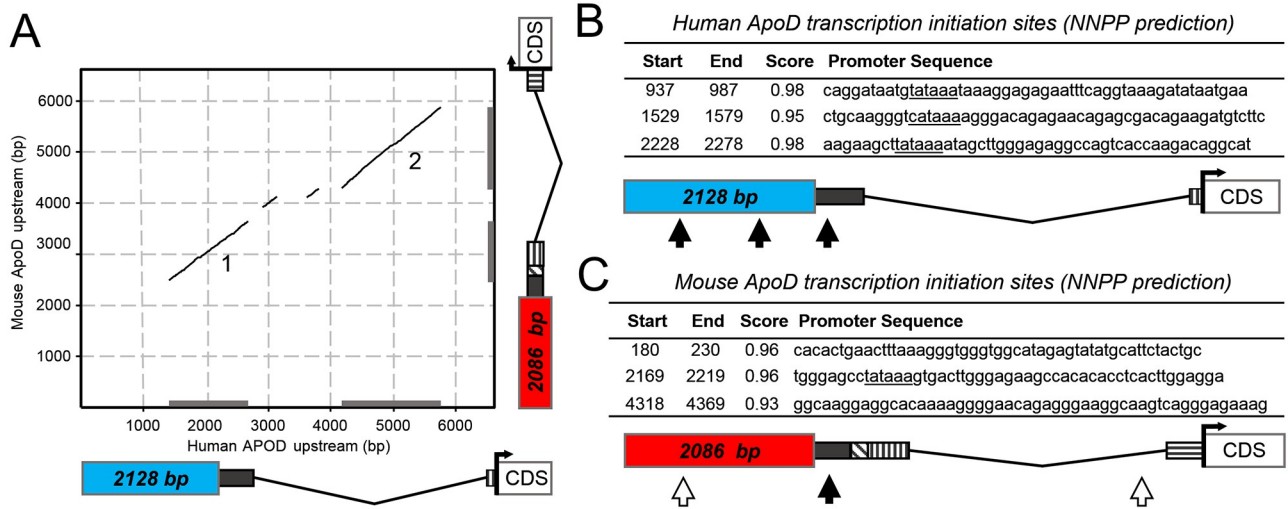

**Fig 5. Promoter predictions for the mouse ApoD gene (I).** A) Dot matrix representation of a pairwise alignment of human vs. mouse ApoD upstream gene sequences, covering approximately 6000 bp upstream of the CDS start site and comprising 5' UTR and the reported/predicted human/mouse promoter regions (colored rectangles). Black and striped boxes indicate ExonScan predicted 5' UTR exons. Region 1 matches a big portion of the reported human promoter and maps to the first mouse 5' UTR exons. Region 2 locates in the 5' UTR long intron in both genes. Only diagonals with p<0.001 are shown. B) The table shows three potential initiation transcription sites in human ApoD locus (black arrows in the diagram) detected by a neural network prediction algorithm. The third one lies within the first 5' UTR exon. Predicted TATA boxes appear underlined. C) Three potential transcription initiation sites are detected in mouse ApoD locus. The second one lies in the 5' UTR, in a region similar to the reported human promoter, and contains a predicted TATA box (black arrow in the diagram). Two other initiation sites (white arrows) do not contain a TATA box, including the site located in the large intron.

have been demonstrated to regulate ApoD transcription in human cells [55]. Similarly, promoter α shows binding sites for NFAT, which forms a cooperative complex with AP1 and is involved in nervous system development [56], and for CBF-α, which has been found to bind the human ApoD promoter [21]. However, promoter β shows a very different pattern of transcription factor binding sites, being CEBP the most common. CEBP is involved in cell proliferation, growth and differentiation, particularly in the nervous system [57]. Other factors like SMAD, FKHD and SP1 are also predicted to regulate ApoD by binding promoter β, though with a lower number of predicted binding sites. An interesting finding is SP1, which is known to bind GC-rich promoter regions and activate transcription in the absence of TATA box or initiator sites [58], as it is the case in promoter β (Fig 6C).

We thus set to experimentally test the expression capabilities of promoter fragments α and β (Fig 6B) using a Luciferase reporter assay in the IMA2.1 mouse astrocytic cell line. These cells show a low vesicular ApoD expression by immunolabeling under control condition, which increases when challenged with PQ (Fig 7A and 7B). This expression regulation is similar to that found in primary mouse astrocytes [15,59] and in the mouse brain upon similar oxidative stress conditions [14,19]. The Luciferase expression driven by promoter α is higher than that of promoter β under control conditions (Fig 7C). Promoter α region drives an enhanced (~1.5-fold) Luciferase activity in response to PQ (Fig 7D), as expected for the reported stress-regulated expression of ApoD, but this response was dampened with increasing PQ concentration. However, PQ triggers in promoter β a 2-fold increase in expression that keeps rising (3-fold) with a higher dose of PQ. These results demonstrate that the promoter β genomic region has the capability to promote expression proportionally to oxidative stress levels, and thus might function as an alternative promoter for ApoD expression in mouse astrocytes. Since this genomic region lies right upstream of the 5' UTR exon 4 sequence that constitutes

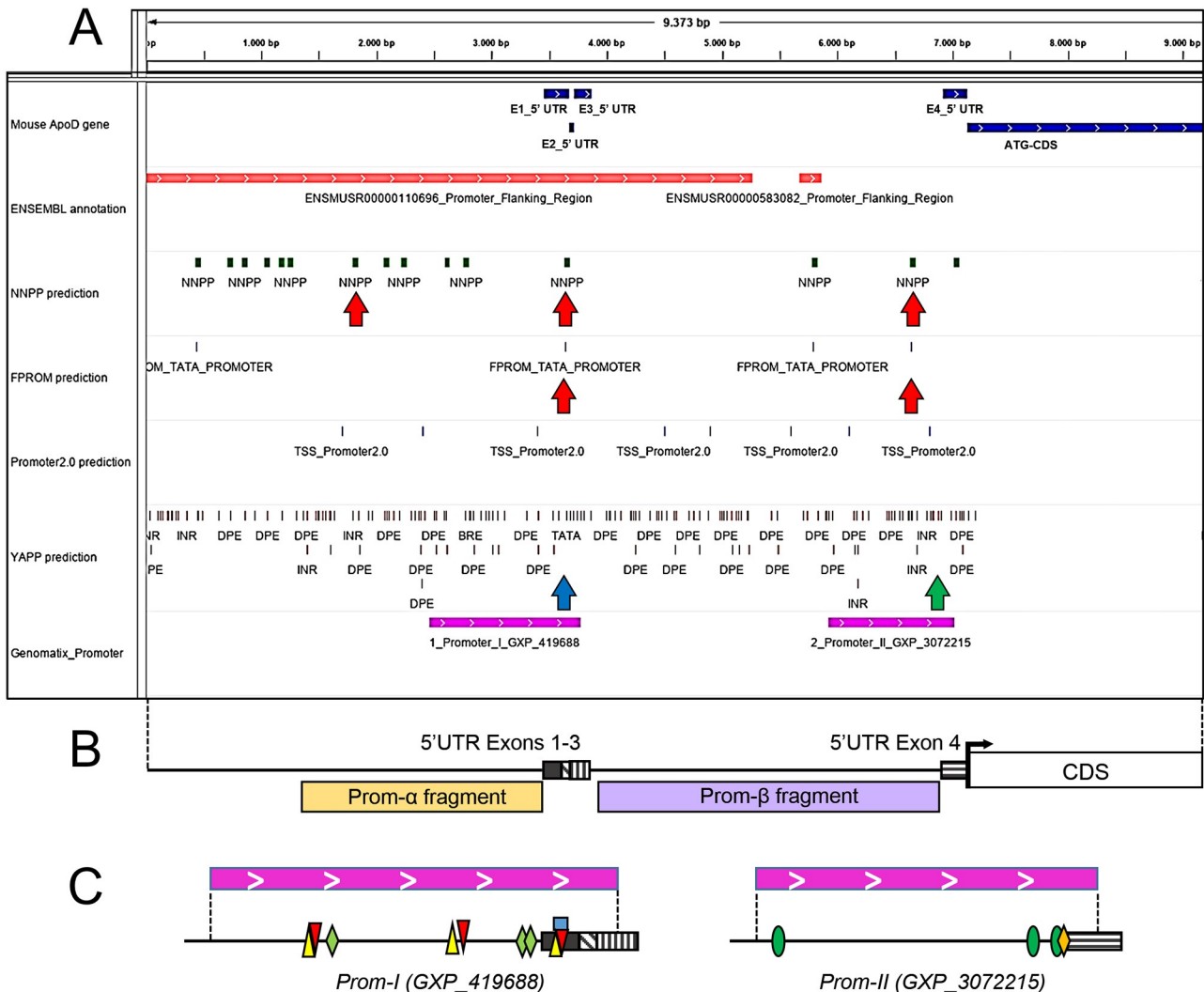

**Fig 6. Promoter predictions for the mouse ApoD gene (II).** A) Bioinformatic predictions of promoter elements in the upstream 5' region of the mouse ApoD coding sequence. Information obtained from the NNPP, FPROM, YAPP and PROMOTER2_0 are visualized with the Integrated Genomics Viewer (IGV). Three high score NNPP-predicted and two FPROM-predicted initiation sites are noted by red arrows. Blue arrow points to the TATA box-containing initiation site in Promoter I (canonical α), as predicted by YAPP. Green arrow points to predicted initiator (INR) and downstream promoter element (DPE) in Promoter II (alternative β) optimally spaced (20–30 bp) for transcription initiation upstream of the 5' UTR exon 4. Purple bars delimit the promoter regions identified by Genomatix. B) Schematic representation of the mouse ApoD locus drawn at the same scale as panel A. Promoter fragments α (2 Kb) and β (3 Kb) cloned for Luciferase assays are represented. They include the regions identified by Genomatix (GXP) as Promoter I and Promoter II respectively. C) Transcription factor binding sites predicted by ModelInspector are mapped onto the GXP promoter regions (enlarged scale). NFKB (yellow triangles), NFAT (red triangles), CBFα (green diamonds), AP1 (blue rectangle), CEBP (green ovals), SP1 (orange diamond). Binding of multiple factors is predicted at various locations.

variant E of ApoD mRNA, its promoter activity could be related to the OS-dependent expression of the 5' UTR variant E.

## Discussion

Apolipoprotein D is one of the few Lipocalins prominently expressed in nervous system. It is characterized by diverse and unique functional traits conditioning nervous system development, maintenance through aging, and response to disease [13,14,15,16,17,18,59,60,61,62].

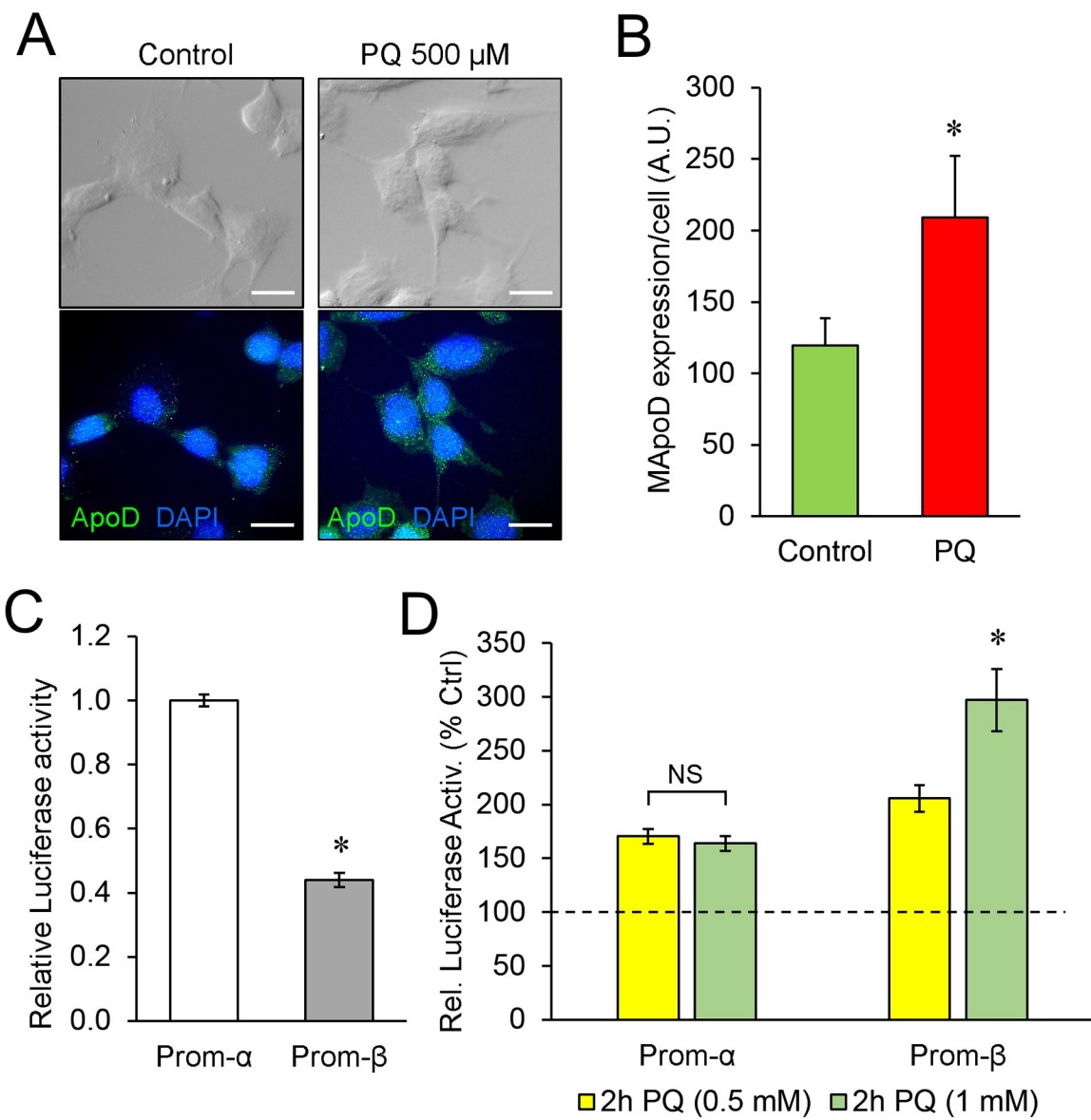

**Fig 7. Canonical and alternative mouse ApoD promoter-driven expression in astrocytes upon PQ treatment.** A) Representative images of differential interference contrast (upper panels) and fluorescence ApoD immunodetection (lower panels) in IMA2.1 mouse astrocyte cells under control and PQ-induced oxidative stress conditions. ApoD expression increases in the astrocytic cell line upon oxidative stress. Calibration bars: 20 μm. B) Mouse ApoD expression evaluated by immunofluorescence signal intensity in IMA2.1 cells in control (48 cells) and PQ (60 cells) conditions. Significance was assessed by unpaired Student's t-test. C-D) Luciferase expression assays in IMA2.1 co-transfected cells, normalized to control. Normalized average values and standard deviation are represented. Expression driven by the Prom-α fragment is constitutively higher than that of Prom-β. Significance was assessed by a Welch's unequal variances t-test. (C). An oxidative stress insult with PQ increases Luciferase activity levels through both promoter constructs, but the highest response at the concentrations tested is driven by Prom-β. Significance was assessed by a two-way ANOVA with Holm-Sidak post-hoc correction. (D).

Regulation of ApoD gene transcription by oxidative stress and metabolic stimuli has been reported, revealing ApoD as a downstream target of JNK and MEK/ERK pathways [15,21]. However, other levels of regulation of gene expression have not been explored.

Translation efficiency has been reported to correlate to the evolutionary history in Lipocalins, mainly due to postranscriptional regulation mechanisms related to the 5' UTR [4].

ApoD makes a particularly attractive case for an in depth study of this region, as this gene is considered the earliest diverging member of the family in chordates [5,63,64], in contrast with the Lipocalins Ptgds and Lcn2, also expressed in the brain, that diverged from progressively more recent family branches. At the sequence level, ApoD 5' UTR shows the signs of strong selective pressure, with high conservation between mammalian orthologs (78.2%). This conservation is in the range of CDS third codon position, compared to 65,7% for Ptgds and 47,1% for Lcn2 5' UTRs [4]. Noteworthy, mouse ApoD stands out among mammalian Lipocalin genes for having the largest number of exons (four) in its 5' UTR, which can splice totally or partially to form up to five alternative variants. This scenario contrasts with a single exon, with two alternative 5' UTRs, in Ptgds, and one exon and single 5' UTR in Lcn2. Our results demonstrate that the potential variability derived from the ability to generate up to five mRNA species with different 5' UTRs is in fact exploited by the mouse ApoD gene. The 5' UTR variants are differentially expressed across tissues and conditions. Three variants (B, D and E) are enriched particularly in the nervous system, with expression levels varying across developmental stages. Previously unexpected but not surprising, given the biological conditions where ApoD overexpression has been reported in the nervous system, one variant (E) shows a strong induction upon oxidative stress in adult mice cerebellum. In contrast to Variant A, which does not have linear elements known to downregulate translation (uORFs and uAUGs) [25], Variants B to E present one or two of these elements, predicted to locate in hairpin secondary structures. This suggests that mRNAs with these 5' UTR variants exert a tight control on ApoD gene translation, explaining the tissue and physiological expression differences.

Strikingly, a comparison of the upstream regions of the human and mouse ApoD orthologs has revealed an additional layer of transcriptional regulation in the form of a functional alternative promoter located in a large 5' UTR intron of the murine gene. This Promoter β (in contrast to the canonical α) lacks typical elements such as TATA box, but instead possesses binding sites for SP1 factor, known to bind GC-rich regions and activate transcription in the absence of TATA box or other initiators [58]. Although Promoter β yields less gene expression than α in cultured mouse astrocytes in control conditions, incremental doses of the oxidative stress generator PQ produce a modulated increase in Promoter β activity that is not present in α-driven expression. We postulate that Promoter β might be working in association with 5' UTR variant E, as a regulatory tandem triggered by oxidative stress that works both at pre and postranscriptional levels.

The wealth of knowledge on ApoD response to brain aging, disease and injury strongly supports that ApoD gene upregulation is an endogenous mechanism of neuroprotection. In this sense, the 5' UTR variation discovered in mouse ApoD should be further explored in the human gene, given its potential role in regulating ApoD response to aging and many neurodegenerative diseases, both causally related to oxidative stress. While protein-based therapies are also possible, pharmacological manipulations of regulators upstream of ApoD endogenous expression should open new therapeutic avenues. The unexpected pattern of 5' UTR variants expression, combined with activation of alternative promoters with different transcriptional regulatory elements, brings a new layer of regulatory complexity worth exploring to understand and manipulate the neuroprotective potential of ApoD.

## Supporting information

**S1 Raw images.**
(PDF)

## Author Contributions

**Conceptualization:** Diego Sanchez, Gabriel Gutierrez, Maria D. Ganfornina.

**Data curation:** Andres Mejias, Gabriel Gutierrez.

**Formal analysis:** Sergio Diez-Hermano, Andres Mejias.

**Funding acquisition:** Diego Sanchez, Maria D. Ganfornina.

**Investigation:** Sergio Diez-Hermano, Andres Mejias, Diego Sanchez.

**Methodology:** Sergio Diez-Hermano, Andres Mejias, Gabriel Gutierrez.

**Supervision:** Diego Sanchez, Gabriel Gutierrez, Maria D. Ganfornina.

**Visualization:** Sergio Diez-Hermano.

**Writing – original draft:** Sergio Diez-Hermano, Diego Sanchez, Maria D. Ganfornina.

**Writing – review & editing:** Andres Mejias, Diego Sanchez, Gabriel Gutierrez, Maria D. Ganfornina.

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
