## [Decision Letter · Decision Letter 0]

8 Apr 2020

PONE-D-19-34292

Control of the neuroprotective Lipocalin Apolipoprotein D expression by alternative promoter regions and differentially expressed mRNA 5’ UTR variants

PLOS ONE

Dear Dr. Ganfornina,

Thank you for submitting your manuscript to PLOS ONE. After careful consideration, we feel that it has merit but does not fully meet PLOS ONE’s publication criteria as it currently stands. Therefore, we invite you to submit a revised version of the manuscript that addresses the points raised during the review process.

We would appreciate receiving your revised manuscript by May 23 2020 11:59PM. To enhance the reproducibility of your results, we recommend that if applicable you deposit your laboratory protocols in protocols.io, where a protocol can be assigned its own identifier (DOI) such that it can be cited independently in the future. For instructions see: http://journals.plos.org/plosone/s/submission-guidelines#loc-laboratory-protocols

We look forward to receiving your revised manuscript.

Kind regards,

Hiroyoshi Ariga

Academic Editor

PLOS ONE

Journal Requirements:

Reviewers' comments:

Reviewer's Responses to Questions

**Comments to the Author**

1. Is the manuscript technically sound, and do the data support the conclusions?

Reviewer #1: Partly

Reviewer #2: Yes

2. Has the statistical analysis been performed appropriately and rigorously? 

Reviewer #1: I Don't Know

Reviewer #2: I Don't Know

3. Have the authors made all data underlying the findings in their manuscript fully available?

Reviewer #1: Yes

Reviewer #2: Yes

4. Is the manuscript presented in an intelligible fashion and written in standard English?

Reviewer #1: Yes

Reviewer #2: Yes

5. Review Comments to the Author

Reviewer #1: Ganfornina et al. describe the potential role of differentially expressed APOD variants in the brain, especially in response to oxidative stress. The authors primarily utilize in silico methods for this work and assess some of their predictions using mice. The authors argue that evolutionary constraints in the 5’ UTR of ApoD suggest an underlying function and show that various isoforms are in fact expressed in different tissues and under oxidative stress. They use in silico approaches to map the ApoD 5’ UTR and predict its secondary structure and promoter. They conclude their study by demonstrating that oxidative stress alters the promoter used to drive expression of ApoD in cultured mouse astrocytes.

While the authors’ findings are interesting, this work is largely preliminary and would necessitate the addition of functional data prior to publication to provide support for their predictions. The majority of the evidence for the author’s predictions is gathered using in silico methods (Figs. 1-2, 4-6) with additional evidence representing limited confirmation of these predictions in vitro (Figs. 3, 7). The computational methodologies used by the authors are not novel and are similar to those used in the author’s previous publication last year in this same journal. Thus, only the in vitro assays represent novel data that isn’t readily available using online tools.

Even though the in vitro data reveal interesting findings, there are many questions remaining as to how ApoD expression is regulated. For the predicted structures, further study is needed to validate predictions (gel electrophoresis or mass spec techniques). Moreover, the authors only probed embryonic whole brain and postnatal cerebellum of mice. As the authors argue that these findings may impact knowledge on the aging brain, it would be important to test postnatal tissue including additional regions of the brain, especially those regions most affected in disease (e.g. Alzheimer’s). Why is the cerebellum preferentially responsive to oxidative stress? Why did the authors choose to examine their mechanism in cultured astrocytes versus other cell types in the brain (Fig. 7)? Is ApoD regulation present in neurons or other glia subtypes? What are the consequences of activating the E variant of ApoD in the brain upon oxidative stress? What impact does the E variant have on ApoD function? Is this an active mechanism to protect the brain from oxidative stress or just a marker of stress? Is this expression difference specific to paraquat or is the same response elicited using other forms of oxidative stress induction? Other phenomena were not investigated or discussed by the authors. For example, the restricted expression of the B variant in the brain suggests specialization of this variant and could be probed further.

In conclusion, the current manuscript is largely based on computational predictions that lack sufficient mechanistic confirmation of the regulation of ApoD expression and, thus, should not be accepted for publication.

Reviewer #2: Evaluation of manuscript PONE-D-19-34292- Diez-Hermano et al 2020.

The manuscript by Diez-Hermano et al. report on the first analysis of the mouse ApoD 5’UTR region. The authors describe 5 novel ApoD variants which are differentially expressed in various tissues with variants B, D, and E showing CNS specificity. The authors further report two novel alternative promoters designated promoter alpha and beta that drive ApoD expression under control and paraquat-generated oxidative stress conditions.

The manuscript is well organized and easy to understand and presents interesting findings. However, there are minor issues that need to be addressed.

First, it is not possible to assess whether the statistical analysis been performed appropriately and rigorously as they are not described in the figure legends although the methods section states that “The tests used for each experiment are described in figure legends”

The reference list should be expanded. This research team has done extensive work on this subject in the past and it is thus understandable that many of the papers cited are from their group. However, this reviewer suggests adding key references on ApoD function in the brain such as He et al 2009; Kim et al 2009; Bhatia et al 2012, etc. The authors could also complement their list of references on ApoD involvement in Alzheimer’s disease with Terrisse et al. J Neurochem. 1998; Belloir et al 2001; Desai et al 2005; Kalman et al. Neurol Res. 2000, etc.

Figure 2 describing the genomic structure of the mouse ApoD 5’ UTR region and predicted mRNA transcripts variants would benefit from adding numbers both as presented in the tables (Fig.2 B and C) as well as in relation to the ORF (negative numbering). This would allow a better understanding of this region in relation to the promoter sequence previously published. The Fig. 2 legend should be extended to include information such as what the arrows in Fig. 2A correspond to.

Data and methods related to the differential expression of each 5’UTR variant are highly confusing and need major revision. The authors state that “RNA obtained from individual samples of the same tissue or experimental condition were pooled in equimolar amounts to be reversed transcribed.” Perhaps they should expand on why such method was used instead of doing the RT on each individual sample. How can they be sure that the quantification is not biased by contaminating residual DNA or degradation sub-products? It is also confusing why the authors limit their investigation in brain to the cerebellum. Given the expression and importance of apoD in neurodegenerative conditions such as Alzheimer’s and schizophrenia, it would be important to investigate the expression of the 5’UTR variants in other brain regions including the prefrontal cortex and hippocampus. Moreover, the primer combination used for each panel of Fig. 3 should be described either in the figure caption or in the methods section. Likewise, Table 1 should be revised. For example, primer pU1-F likely was used to amplify variants A, C and D but is assigned as used in “genomic PCR” only. Location (number) of the primers would be helpful. The difference between Fig. 3C and 3D must be explained. Were different primers used? In which case they should appear in Table 1. Lastly, since the authors speak of differential expression between tissues and conditions (lines 273-277; Fig. 3 C-F), the expression of a housekeeping gene should also be presented.

Regarding Fig. 7, a quantification of the increase in ApoD levels in IMA2.1 mouse astrocyte cells in control compared to OS conditions would be most informative. It would have also been interesting to include the full promoter region (represented in Fig. 6B) in the luciferase assays (Fig. 7) to better evaluate the extent of the effects that are specific to promoter alpha and beta under control and oxidative stress conditions.

In the discussion, the authors could further elaborate on the potential functional roles of the ApoD variants in control and in disease conditions, in particular oxidative stress conditions.

6. PLOS authors have the option to publish the peer review history of their article (what does this mean?). If published, this will include your full peer review and any attached files.

Reviewer #1: No

Reviewer #2: No

---

## [Author Response · Author response to Decision Letter 0]

15 May 2020

Reviewer 1.

We are glad the Reviewer considers our in silico and experimental data and findings interesting. The Reviewer raises some general considerations on the completion of our study, which we are addressing below.

1) “The computational methodologies used by the authors are not novel and are similar to those used in the author’s previous publication…”

The bioinformatics analysis we have performed is not “novel” (methodologically speaking, as we have not developed any new algorithm to study the ApoD upstream gene regions), but we fail to see this kind of “novelty” as a requisite in the publication criteria of PLoS ONE. In addition, we have used different “computational methodologies” from those used to study the Lipocalin family in our previous publication to study ApoD promoter and to design experimental molecular biology tools.

2) “…only the in vitro assays represent novel data that isn’t readily available using online tools”

No wonder online tools can be “readily available” for everyone. However, only if we use them to uncover interesting predictions that can be tested experimentally, they do answer pertinent questions and contribute to Science advancement. This is the case for the interesting role of upstream UTRs in Lipocalins, and the reviewers judging those in fully silico data of our previous publication raised no concern. We believe this also applies to data included in the current manuscript: ApoD mRNA has been studied for a long time as if it were a single molecular species, without noticing that an alternative promoter and diverse 5’UTR versions could condition its expression. In this work we uncover predictions and test them experimentally, which is per se novel.

3) “…the authors only probed embryonic whole brain and postnatal cerebellum of mice.”

We think the reviewer misread the information mentioned in the Methods section and shown in Fig. 3, where we describe that our PCR experiments were also performed in cerebella of adults (6 months old) mice, both control and subjected to experimental oxidative stress.

4) “Why is the cerebellum preferentially responsive to oxidative stress?

It was stated in line 277 of our manuscript, “Cerebellum was selected because of its reported expression of ApoD and its sensitivity to PQ [19]”. We have added more information in the introduction section (lines 71-73) to help the reader better understand this choice from the beginning.

5) “Why did the authors choose to examine their mechanism in cultured astrocytes versus other cell types in the brain…?”

We selected astrocyte cultures because it is one of the cell types expressing ApoD, where we have in depth knowledge about the role of ApoD in oxidative stress-dependent autocrine and paracrine protection. We explain in our manuscript (lines 435-438), that astrocytes “…show a low vesicular ApoD expression by immunolabeling under control condition, which increases when challenged with PQ (Figure 7A). This expression regulation is similar to that found in primary mouse astrocytes [14,54] and in the mouse brain upon similar oxidative stress conditions [13,18].” 

6) Statistical analysis.

In the revised version of the manuscript, we have included a more detailed explanation of the statistical tests used in our experiments.

In a more general sense, the Reviewer poses many interesting scientific questions not addressed in this work and concludes that much work must be done before we can publish it.

Although we agree that a more detailed study could reveal more mechanistic insights and details into ApoD function regulation, we would like the Reviewer to view our data from a different perspective.

Our laboratory works with model organisms (Drosophila and mouse) on the physiology of ApoD, but is not currently funded to study in depth gene transcriptional regulation or UTR secondary structures. However, after our last publication we decided to explore ApoD 5’ UTR regions as a side project to our main grant objectives. We were able to demonstrate with standard molecular biology techniques that the predicted alternative 5’ UTRs and a novel promoter do in fact exist, and could underlie differential expression in mouse tissues and in response to oxidative stress challenges. An interesting finding acknowledged by the Reviewer.

We agree that (as always in Science) more work can be done, and would actually love to answer all questions related to this interesting topic raised by the Reviewer. However, we are also aware that our current situation prevents to keep developing that project. That is why we believe it is important to disseminate our results to the scientific community interested in transcriptional regulation, instead of hiding them in our desks waiting for a major shift in our lab goals and funding opportunities. We sincerely predict that this work would spawn future research in the field of gene expression regulation.

 

Reviewer 2.

We are glad the Reviewer considers our findings interesting, and want to thank his/her careful and useful review. We address the Reviewer’s concerns below and in the revised version of the manuscript.

1) Statistical analysis description.

A detailed explanation of the statistical tests used in our experiments are included in the revised version of the manuscript.

2) References missing in the general description of ApoD function in the brain.

Thanks for the suggestion. We have included the suggested references in the revised manuscript.

3) Figure 2 data and design.

Thanks for the idea. We have reorganized and expand the figure panels and legends according to the Reviewer’s suggestions.

4) Data and methods related to the differential expression of each 5’UTR variant.

We have revised and rewritten this section taking into account the Reviewer’s concerns. Below are our answers to these questions.

- The primer combination used for each panel of Fig. 3 should be described.

We have included the primer combination used in each panel of the figure. Thanks for the suggestion.

- The difference between Fig. 3C and 3D must be explained. Were different primers used?

Yes, those amplifications were carried out with different sets of primers. We had to use primers pU1F or pU2F to be able to separate variant A from C expression (several variants showed overlapping size). Variants B, D and E were easily identified with the primer pairs cited in the figure.

- Table 1 should be revised.

We have done so, and the reviewer is right about primer pU1-F and its use in RT-PCR. We have checked and corrected the experimental use of all primers listed in the revised table.

- RNA individual samples vs. Sample pools:

The Reviewer is right that two options can be used to account for biological and technical variabilities in experimental design. Biological variability can be evaluated by using individual samples for PCR amplifications, and technical variability would be evaluated by replicating amplifications. An alternative strategy could be to homogeneously mix equimolar amounts of RNA extracted from each individual tissue sample (obviously checking for RNA purity and integrity), thus rendering each tissue in a single pooled sample. Technical variability can be subsequently tested by experiment replicas.

We believe this approach fulfills an adequate assessment of both experimental variabilities present in biological research, and adds an economical aspect to consider (both in personal and reagent lab expenditure) for a scientist. In fact, we have been using this strategy in previous publications from our lab (Ganfornina et al 2005; Bajo et al., 2011; Sanchez et al., 2015) reporting expression data from microarray experiments, where resources and reagents costs are an issue to consider.

Moreover, the sample pooling strategy allows us to show 8 RTs (8 lanes) per PCR shown in figure 3B-F, instead of 42 RTs (and corresponding lanes) from individual sample experiments.

- Quantification bias by contaminating residual DNA or degradation sub-products

RNA purity and integrity were monitored by 260/230 and 260/280 ratios measured with a spectrophotometer, and by agarose electrophoresis. Putatively contaminating DNA was avoided by DNAse treatment, and confirmed by RT(-) amplifications. These details have been added to the revised Methods section.

- Why the authors limit their investigation in brain to the cerebellum?

As explained in the manuscript (line 277) the cerebellum was selected because of its reported expression of ApoD and its sensitivity to PQ [18]. We have added more information in the introduction section to help the reader understand this choice (lines 71-73). We agree that studying the role of 5’ UTR regulation of ApoD in neurodegenerative conditions will be very interesting, but we chose to test for the relevance of 5’ UTR variants in an experimental oxidative stress model well studied and reliably reproduced in our lab.

- Differential expression between tissues and conditions and housekeeping gene.

We did not include a housekeeping gene in those semiquantitative PCR experiments because we wanted to compare the expression of the 5’ UTR variants with that of ApoD CDS (shown in Fig. 3B), which in this case acts as the reference. We agree that when we refer to “differential expression of each 5’ UTR variant” is not referred to a housekeeping gene, and therefore we have rephrased the paragraph in the manuscript stating the comparison with ApoD CDS in each tissue/condition.

In quantitative real-time PCR (Fig. 3G,H), though, we used Rpl18 as a housekeeping gene.

- Quantification of the increase in ApoD levels in IMA2.1 mouse astrocyte cells in control compared to OS conditions.

We have performed the experiments suggested by the Reviewer, measuring ApoD expression by fluorescent immunocytochemistry and confirming the PQ-dependent overexpression. The new data was added as a new panel to Figure 7 (panel B).

- To include full promoter region in the luciferase assays.

We agree with the reviewer that this should be something to pursue. However, we had trouble cloning the whole DNA piece into our vectors, and proceeded to study them as separate entities.

- Further discussion on the potential roles of the ApoD variants in control and in disease conditions.

Although it is tempting to discuss the role of ApoD variants in human diseases, we feel our findings should be considered with caution, given their study in rodents. Thus, we have included a short statement in the revised version of the manuscript prompting the ApoD community to check the putative existence of 5’ UTR variation for the human protein, and its role in human diseases.

---

## [Decision Letter · Decision Letter 1]

29 May 2020

PONE-D-19-34292R1

Control of the neuroprotective Lipocalin Apolipoprotein D expression by alternative promoter regions and differentially expressed mRNA 5’ UTR variants

PLOS ONE

Dear Dr. Ganfornina,

Thank you for submitting your manuscript to PLOS ONE. After careful consideration, we feel that it has merit but does not fully meet PLOS ONE’s publication criteria as it currently stands. Therefore, we invite you to submit a revised version of the manuscript that addresses the points raised during the review process.

We look forward to receiving your revised manuscript.

Kind regards,

Hiroyoshi Ariga

Academic Editor

PLOS ONE

Reviewers' comments:

Reviewer's Responses to Questions

**Comments to the Author**

1. If the authors have adequately addressed your comments raised in a previous round of review and you feel that this manuscript is now acceptable for publication, you may indicate that here to bypass the “Comments to the Author” section, enter your conflict of interest statement in the “Confidential to Editor” section, and submit your "Accept" recommendation.

Reviewer #1: (No Response)

Reviewer #2: All comments have been addressed

2. Is the manuscript technically sound, and do the data support the conclusions?

Reviewer #1: Yes

Reviewer #2: Yes

3. Has the statistical analysis been performed appropriately and rigorously? 

Reviewer #1: Yes

Reviewer #2: Yes

4. Have the authors made all data underlying the findings in their manuscript fully available?

Reviewer #1: Yes

Reviewer #2: Yes

5. Is the manuscript presented in an intelligible fashion and written in standard English?

Reviewer #1: Yes

Reviewer #2: Yes

6. Review Comments to the Author

Reviewer #1: The authors did not adequately address the questions posed in our initial review of this paper and there are still too many unknowns to warrant publication of this manuscript at this time. The editorial decision was for major revision, but only minor edits were made to the manuscript and no additional data was provided. It should be noted that while the figures were reordered, the old figure titles remain, making the reading of the manuscript confusing.

The authors concede that the bioinformatic approach to this problem, which makes up the majority of the data presented herein, is not novel. Moreover, the in vitro assays provided in the paper are inadequate to definitively support the authors conclusions. While the authors state in their response that they have additional data in mice that might bolster their conclusions, they did not include those data. Without the addition of these or other in vivo data, this paper remains inadequate for publication.

Reviewer #2: (No Response)

7. PLOS authors have the option to publish the peer review history of their article (what does this mean?). If published, this will include your full peer review and any attached files.

Reviewer #1: No

Reviewer #2: No

---

## [Author Response · Author response to Decision Letter 1]

3 Jun 2020

We are glad that our responses and changes made to the manuscript addressed all of Reviewer #2 questions, and want to thank the reviewer for the improvements suggested for the revised version of our manuscript.

As for Reviewer #1, we are also glad that he/she accepts our responses to previous questions on experimental details: e.g., why we chose cerebellum as a tissue type, and cultured astrocytes as a cell type to test the main question in this work, and that the bioinformatically predicted 5’ UTR variants and alternative promoter do in fact exist and influence the Lipocalin ApoD gene expression. We are also glad the reviewer's positive marks to all editorial questions in the reviewer form.

Answering to some concerns:

1) …”only minor edits were made to the manuscript and no additional data was provided”.

The revised version of the manuscript includes many changes addressing all of the constructive criticisms posed by Reviewer #2. Moreover, panel B in figure 7 shows new data and analysis from immunocytochemical experiments proposed by Reviewer #2. Thus, and again, we must correct Reviewer #1 in his/her argument that “no additional data was provided”. Moreover, we consider utterly unfair this statement, given the exceptional situation under which we have made those experiments in such difficult times of all our professional lives.

2) …”while the figures were reordered, the old figure titles remain, making the reading of the manuscript confusing.”

As far as we can tell, we have not reordered any figure in this manuscript! We just included amendments suggested by Reviewer #2 to some panels, and the already cited panel B in figure 7 showing the new experimental data required by Reviewer #2. All those changes were incorporated in figure legends, and therefore we see no point in changing “figure titles”. Finally, we do not understand that Reviewer #1 now finds reading the manuscript “confusing” while answering “Yes” to comment 5.

3) …”concede that the bioinformatic approach to this problem, which makes up the majority of the data presented herein, is not novel”.

While maintaining the argument that we do not see “novelty” as a requisite in the publication criteria of PLoS ONE, we see this comment as a rhetorically twisted reading of our response. To his/her initial consideration that “The computational methodologies used by the authors are not novel and are similar to those used in the author’s previous publication “, we answered that we use in this manuscript different computational methodologies to study ApoD promoter and to design experimental molecular biology tools. So, the methodologies are not “similar”. The only “concession” we made is that we did not develop any bioinformatic (novel) algorithm in our analysis. We are not bioinformaticians indeed! However, we must insist in that the bioinformatic approach we have used has discovered novel biological facts related to ApoD function.

4) … “While the authors state in their response that they have additional data in mice that might bolster their conclusions, they did not include those data.”

We cannot find in our response a statement that we have “additional data in mice”. We obviously did not include additional data performed in mice because we do not have them! The data we show in this manuscript are the data we have. We cannot provide “other in vivo data”, as the reviewer requires, because we cannot collect them in the period of a manuscript revision.

---

## [Editor Report · Decision Letter 2]

4 Jun 2020

Control of the neuroprotective Lipocalin Apolipoprotein D expression by alternative promoter regions and differentially expressed mRNA 5’ UTR variants

PONE-D-19-34292R2

Dear Dr. Ganfornina,

We’re pleased to inform you that your manuscript has been judged scientifically suitable for publication and will be formally accepted for publication once it meets all outstanding technical requirements.

Kind regards,

Hiroyoshi Ariga

Academic Editor

PLOS ONE

---

## [Editor Report · Acceptance letter]

10 Jun 2020

PONE-D-19-34292R2 

Control of the neuroprotective Lipocalin Apolipoprotein D expression by alternative promoter regions and differentially expressed mRNA 5’ UTR variants 

Dear Dr. Ganfornina:

I'm pleased to inform you that your manuscript has been deemed suitable for publication in PLOS ONE. Congratulations! Your manuscript is now with our production department. 

Kind regards, 

on behalf of

Dr. Hiroyoshi Ariga 

Academic Editor

PLOS ONE